# RLTHF: Targeted Human Feedback for LLM Alignment

Yifei Xu [1 2 *]   Tusher Chakraborty [1]   Emre Kıcıman [1]   Bibek Aryal [1]   Srinagesh Sharma [1]   Songwu Lu [2]
Ranveer Chandra [1]

## Abstract

Fine-tuning large language models (LLMs) to align with user preferences is challenging due to the high cost of quality human annotations in Reinforcement Learning from Human Feedback (RLHF) and the generalizability limitations of AI Feedback. To address these challenges, we propose RLTHF, a human-AI hybrid framework that combines LLM-based initial alignment with selective human annotations to achieve full-human annotation alignment with minimal effort. RLTHF identifies hard-to-annotate samples mislabeled by LLMs using a reward model's reward distribution and iteratively enhances alignment by integrating strategic human corrections while leveraging LLM's correctly labeled samples. Evaluations on HH-RLHF and TL;DR datasets show that RLTHF reaches full-human annotation-level alignment with only 6-7% of the human annotation effort. Furthermore, models trained on RLTHF's curated datasets for downstream tasks outperform those trained on fully human-annotated datasets, underscoring the effectiveness of RLTHF.

## 1. Introduction

In recent years, large language models (LLMs) have demonstrated remarkable advancements, unlocking new possibilities across a wide range of applications (Touvron et al., 2023; Jiang et al., 2024; Achiam et al., 2023; Team et al., 2023). As these models become more powerful, the focus has shifted toward customization, i.e., fine-tuning base models to better serve specific tasks and user needs (Wei et al., 2021; Li et al., 2023a). Companies are increasingly investing in solutions built upon fine-tuned models, recognizing the value of adapting LLMs to align with end-user preferences, including intent, style, grounding, and compliance requirements (Atreya, 2024; Microsoft, 2024; Sharma, 2024; AI, 2024). A key approach to achieving this alignment is Reinforcement Learning from Human Feedback (RLHF), which has emerged as a widely adopted technique in the literature for refining model behavior based on human feedback (Bai et al., 2022a; Stiennon et al., 2020; Rafailov et al., 2024; Wang et al., 2024a; Ouyang et al., 2022).

The effectiveness of RLHF techniques heavily depends on high-quality human annotations, which are both costly and time-consuming to obtain (Pang et al., 2023; Lee et al., 2023; Wang et al., 2024a). To mitigate this challenge, Reinforcement Learning from AI Feedback (RLAIF) has been introduced, leveraging LLMs to replace human annotators in the feedback loop (Lee et al., 2023; Lee et al.; Bai et al., 2022b). While RLAIF can approximate human judgment to some extent, it is sensitive to factors such as prompt optimization, task complexity, model bias, generator-discriminator gap, and the capability of the judge model, limiting its ability to fully replicate human annotations (Huang et al., 2024; Sharma et al., 2024; Lee et al., 2023; Zeng et al., 2024; Huang et al., 2023). Our evaluation also provides evidence of these limitations. Furthermore, the samples that challenge a judge model are often the ones most critical for adapting base models to specialized fine-tuning tasks (Ethayarajh et al., 2024; Yuan et al., 2024; Huang et al., 2023). The cost of human annotation is further exacerbated by privacy and security constraints that restrict fine-tuning service providers' access to an entire customer data corpus. In such cases, only subject matter experts (SMEs) within the customer organization have full visibility into the data, making it particularly difficult to optimize prompts effectively across the entire corpus, especially for hard-to-annotate samples.

To address these challenges, we propose Reinforcement Learning from Targeted Human Feedback (RLTHF), a human-AI hybrid solution that combines coarse initial alignment using general-purpose LLMs with the progressive integration of strategically selected human annotations to achieve annotation quality comparable to fully human-supervised approaches. RLTHF begins with an initial alignment stage, where a general-purpose LLM labels unlabeled data based on high-level instructions. While this approach effectively captures broader human alignment for easier data

---

*Work done during an internship at Microsoft. [1]Microsoft [2]University of California, Los Angeles. Correspondence to: Yifei Xu <yxu@cs.ucla.edu>, Tusher Chakraborty <tusher.chakraborty@microsoft.com>.

*Proceedings of the 42$^{nd}$ International Conference on Machine Learning*, Vancouver, Canada. PMLR 267, 2025. Copyright 2025 by the author(s).

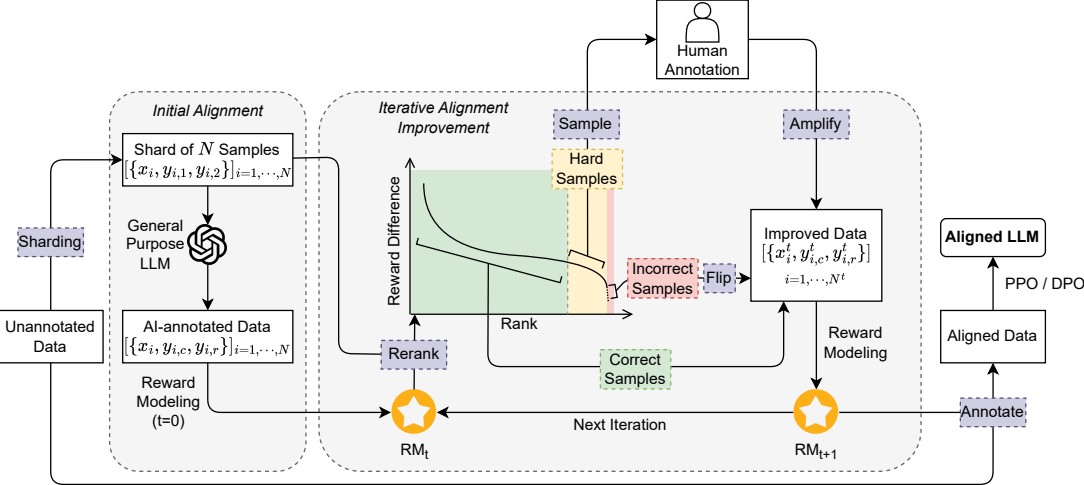

Figure 1: Overview of RLTHF process. RLTHF starts with coarse LLM alignment on the task. It then iteratively takes targeted human feedback and blending the same with sanitized LLM-annotations to reach the complete human alignment, leveraging reward distribution of a reward model in its training dataset.

points, it often struggles with fine-grained nuances, leading to incorrect labeling. RLTHF automatically identifies these hard-to-annotate data points and directs human effort exclusively toward them. This targeted approach enables RLTHF to achieve the quality of fully human-annotated data while reducing the majority of human annotation effort.

To enable this efficient human-in-the-loop approach for achieving comprehensive human alignment, RLTHF introduces the following key technical contributions:

First, we introduce a concept that leverages the reward distribution of a reward model over its training dataset to capture the relative arrangement of samples based on rewarded features. This distributional property enables the identification of both potential annotation inaccuracies and the model's confidence across the dataset. Specifically, we train a reward model on the LLM-labeled dataset to uncover clusters of hard-to-annotate samples that are *highly* likely to be mislabeled by the LLM, as well as easy samples that are *highly* likely to be correctly labeled.

Building on this concept, we propose an innovative iterative reward model training technique to achieve oracle-level human alignment in the dataset. In each iteration, RLTHF identifies highly probable mislabeled data points and rectifies the same using human annotations. Simultaneously, it detects clusters of samples that are very likely to be correctly labeled by the LLM and incorporates them with human-annotated data to construct a high-quality training set for the next iteration of reward model training. Throughout this process, RLTHF preserves data richness and maximizes the efficiency of human annotation investment through carefully controlled hyperparameters.

Finally, we evaluate RLTHF on two distinct preference

datasets: HH-RLHF and TL;DR. Our results demonstrate that RLTHF achieves accuracy comparable to a fully human-annotated dataset while requiring only 6–7% of the total human annotations. Furthermore, we conduct a comparative study by training models on downstream tasks using DPO (Rafailov et al., 2024). Remarkably, models trained with RLTHF even outperform those trained on fully human-annotated datasets, highlighting the impact of RLTHF's meticulous data curation in enhancing model performance.

## 2. Background and Related Work

LLMs have demonstrated impressive performance across a wide spectrum of tasks (Achiam et al., 2023; Dubey et al., 2024; Team et al., 2023). Despite the progress, their performance on customized downstream tasks can be significantly optimized by supervised fine-tuning (SFT) with instruction and human-written responses pairs (Chung et al., 2024; Thoppilan et al., 2022). Reinforcement learning with preference data has further shown success due to the easier-to-collect data form (Ouyang et al., 2022; Stiennon et al., 2020; Lee et al., 2023). Representative methods include Proximal Policy Optimization (PPO) (Schulman et al., 2017), which optimizes the LLM with a separate reward model, and Direct Preference Optimization (DPO) (Rafailov et al., 2024), which directly learns from the preference data. Although an easier data collection is available, these methods still largely rely on the richness and quality of the preference data (Xu et al., 2024; Zheng et al., 2023b; Wang et al., 2024a).

### 2.1. Alignment with External Feedback

Human feedback is regarded as the golden standard in LLM alignment. However, reinforcement learning from human feedback (RLHF) (Ouyang et al., 2022; Stiennon et al.,

2020; Köpf et al., 2024) typically incorporates heavy and expensive crowdsourcing efforts or expert annotations to guarantee data diversity and richness. To relieve the reliance on human effort, reinforcement learning with AI feedback (RLAIF) (Lee et al., 2023; Bai et al., 2022b) provides an alternative that collects feedback from stronger LLMs instead of humans. On the other hand, this method is limited by the capability of the stronger LLM annotators (Huang et al., 2024; Sharma et al., 2024; Lee et al., 2023) especially for customized tasks, and suffers from their intrinsic biases (Zheng et al., 2023a). In this paper, we take advantage of RLAIF to establish an initial alignment and strategically incorporate human feedback to efficiently bring LLMs to the true alignment.

## 2.2. LLM Self-Improvement

To break the upper bound of LLMs, recent efforts have been devoted to enabling LLMs to self-improve. Self-Rewarding LMs (Yuan et al., 2024) and Math-shepherd (Wang et al., 2024b) demonstrate the possibility of LLM self-improvement with reward signals from itself. SELF-ALIGN (Sun et al., 2024b) uses a carefully written set of principles to guide LLMs through self-improvement. SER (Huang et al., 2024) starts with only a fraction of human annotations to achieve full-annotation performance by progressively generating additional training data for itself. However, these methods still suffer from the intrinsic upper bound of LLMs and self-improvement is not guaranteed for customized tasks. RLTHF, on the other hand, efficiently introduces human intelligence into the improvement process, thereby ensuring that the improvement is not bounded by LLMs' initial lack of domain understanding.

## 3. Improving Human Alignment with RLTHF

RLTHF enhances alignment with human in preference datasets used for training preference optimization techniques like DPO and PPO. It facilitates LLM training for various downstream tasks, including summarization, compliance, and grounding. Starting with an unlabeled preference dataset, RLTHF strategically integrates AI-generated labels with selective human feedback to maximize alignment while minimizing annotation effort. As illustrated in Figure 1, RLTHF operates in three stages: 1) *Initial alignment*, where an off-the-shelf LLM provides dataset labeling to establish a coarse task understanding, 2) *Iterative alignment improvement*, which leverages reward distribution by a reward model (RM) to locate hard-to-annotate samples mislabeled by the LLM and rectify with selective human feedback while investing the correct LLM labels, 3) *Transferring knowledge for downstream task*, where the curated preference dataset is fed into the DPO pipeline or the trained RLTHF reward model is integrated into the PPO pipeline. Find the corresponding pseudocode in Appendix B.

## 3.1. Initial Alignment

This stage aims to establish an initial coarse alignment in the unlabeled dataset using a general-purpose LLM, which provides preference annotations for each unannotated sample. Prior research suggests that model selection here depends on task complexity relative to the model's capability (Snell et al., 2024). While RLTHF is not found to be sensitive to the choice of model at this stage, a well-suited model can accelerate alignment convergence. The only assumption is that the general-purpose LLM possesses a basic understanding of the downstream task, enabling it to provide a rough initial alignment that serves as a seed for RLTHF.

Our prompt for obtaining preference judgments from the LLM consists of three components: 1) task description, 2) preference judgment principles provided by the user, and 3) few-shot examples with optional chain-of-thought reasoning. The prompt templates are detailed in Appendix A. We do not perform explicit fine-grained prompt tuning, as full visibility into the data may be restricted when offering fine-tuning services to third-party customers. However, to ensure that the selected LLM with our prompt attains a rough level of alignment, we perform an eyes-off validation using strategic human feedback, as detailed in Section 3.2.3.

As mentioned earlier, this AI-generated feedback is prone to errors due to factors such as model biases from pre-training data, task complexity, and prompt optimization, which is also evident in our evaluation. When our ultimate goal is to customize an existing model through fine-tuning to align with end-user preferences, we inherently assume that an off-the-shelf LLM lacks comprehensive alignment with the end-user. However, RLTHF builds upon the initial AI-provided alignment and systematically refines it in subsequent stages to achieve oracle-level human alignment.

### 3.2. Iterative Alignment Improvement

In this stage, we refine the LLM-labeled preference dataset by iteratively training an RM with selective human annotations to enhance alignment. Before diving into the details of this process, we first establish the premise for RM.

#### 3.2.1. REWARD MODEL

Given a labeled preference dataset $\mathcal{D}_{\mathbf{\Lambda}} = \{x_i, y_{i,c}, y_{i,r}\}$, where $i \in [N]$, $x_i$ is the prompt, $y_{i,c}$ and $y_{i,r}$ denote the chosen and rejected completions, respectively, as labeled according to the annotator's preference, $\mathbf{\Lambda}$. Here, if we represent the relative preference orientation of $i^{th}$ completion pair with $\lambda = [-1, +1]$, $\mathbf{\Lambda}$ is a $N$-dimensional vector consists of $[\lambda_i]_{i=1}^N$, meaning that flipping the preferences of all completion pairs results in $\mathcal{D}_{-\mathbf{\Lambda}}$. To train an RM on this dataset, we can formulate the probability distribution of $y_{i,c}$ being preferred over $y_{i,r}$ given $x_i$ as an input, following the

Bradley-Terry (BT) model (David, 1963).

$$P(x \succ y) = \sigma(r(x_i, y_{i,c}) - r(x_i, y_{i,r})) \qquad (1)$$

where $\sigma(\cdot)$ denotes the sigmoid function and $r(\cdot)$ denotes the reward function. Assuming the existence of a true deterministic reward function, the goal is to train the RM to learn this function and predict the reward, $\hat{r}(x, y)$. The RM training can be framed as a binary classification problem (Sun et al., 2024a), where a labeled pair of $\rho_{i,c} := (x_i, y_{i,c})$ and $\rho_{i,r} := (x_i, y_{i,r})$ is passed to the model to predict the conditional class probability according to Eq. 1. This leads to the negative log-likelihood loss function for training.

$$\mathcal{L}(\hat{r}) = -\mathbb{E}_{(x,y)\sim\mathcal{D}}[\log \sigma(\hat{r}(\rho_{i,c}) - \hat{r}(\rho_{i,r}))] \qquad (2)$$

In essence, during the RM training, we pass a preference pair $\{\rho_{i,c}, \rho_{i,r}\}$ labeled as $\rho_{i,c}$ winning over $\rho_{i,r}$ according to the annotator's preference $\mathbf{\Lambda}$. Provided sufficient preference samples in a dataset, the RM learns the winning preference features of the data that determine the winner in a pair, captured in the reward function $\hat{r}_{\mathbf{\Lambda}}$.

### 3.2.2. LOOKING AT REWARD DISTRIBUTION

At this stage, we analyze the distribution of the predicted reward function ($\hat{r}_{\mathbf{\Lambda}}$) within the training preference dataset $\mathcal{D}_{\mathbf{\Lambda}}$. For each labeled preference pair $\{\rho_{i,c}, \rho_{i,r}\}$, we compute the reward score difference as $\Delta_{\mathbf{\Lambda}}\hat{r}_{\mathbf{\Lambda}} = (\hat{r}_{\mathbf{\Lambda}}(\rho_c) - \hat{r}_{\mathbf{\Lambda}}(\rho_r))$. It is important to note that $\Delta_{\mathbf{\Lambda}}$ quantifies the relative preference score of a given pair in alignment with the annotator's preference orientation $\mathbf{\Lambda}$, satisfying the property $\Delta_{\mathbf{\Lambda}}r = -\Delta_{-\mathbf{\Lambda}}r$. By ranking all preference pairs in $\mathcal{D}_{\mathbf{\Lambda}}$ based on $\Delta_{\mathbf{\Lambda}}\hat{r}_{\mathbf{\Lambda}}$, a monotonic reward distribution curve, denoted as $\vartheta(\Delta_{\mathbf{\Lambda}}\hat{r}_{\mathbf{\Lambda}})$, emerges. This distribution, as depicted in Figure 2a, provides insight into the model's reward assignment across the dataset, though for the moment, the legend in the graph can be disregarded.

The reward distribution curve $\vartheta(\Delta_{\mathbf{\Lambda}}\hat{r}_{\mathbf{\Lambda}})$, derived from the training preference dataset $\mathcal{D}_{\mathbf{\Lambda}}$, reflects the degree of alignment the RM (trained with optimal validation loss) has achieved across $\mathcal{D}_{\mathbf{\Lambda}}$ during training. The upper left region of the curve consists of samples with high positive $\Delta_{\mathbf{\Lambda}}\hat{r}_{\mathbf{\Lambda}}$, indicating strong agreement between the RM and the training preference labels $\mathbf{\Lambda}$. This suggests that the RM effectively identifies and reinforces strong winning preference features in these samples, implying that these features were dominant in $\mathcal{D}_{\mathbf{\Lambda}}$. Conversely, the bottom right region of the curve contains samples with very low or even negative $\Delta_{\mathbf{\Lambda}}\hat{r}_{\mathbf{\Lambda}}$, signaling disagreement between the trained reward function $\hat{r}_{\mathbf{\Lambda}}$ and the training preference labels for these samples. This misalignment arises from two primary factors. (1) Absence of strong features, where RM is not able to find any strong preference feature in these samples according to $\hat{r}_{\mathbf{\Lambda}}$. (2) Conflicting samples within $\mathcal{D}_{\mathbf{\Lambda}}$, where the preference

features of these samples are highly conflicting with other stronger preference features learned in $\hat{r}_{\mathbf{\Lambda}}$, leading the RM to penalize them.

### 3.2.3. RLTHF LEVERAGING REWARD DISTRIBUTION

RLTHF trains the initial RM using a preference dataset labeled by a general-purpose LLM from the previous stage. We denote this dataset as $\mathcal{D}_{\mathbf{\Lambda}_{\mathbf{LLM}}}$ where $\mathbf{\Lambda}_{LLM}$ represents the LLM's preference labeling. Since the RM training includes a validation set derived from $\mathcal{D}_{\mathbf{\Lambda}_{\mathbf{LLM}}}$, this ensures that the trained RM is broadly aligned with the LLM's preferences. We assume that the LLM has a coarse but reasonable understanding of preference judgments, particularly for relatively easy-to-annotate samples. As a result, the features of these samples dominate in $\hat{r}_{\mathbf{\Lambda}_{\mathbf{LLM}}}$. Based on our earlier discussion, the upper left region of the reward density curve, $\vartheta(\Delta_{\mathbf{\Lambda}_{\mathbf{LLM}}}\hat{r}_{\mathbf{\Lambda}_{\mathbf{LLM}}})$ contains high density of samples with prominent preference features, i.e., those that are easier for the LLM to annotate accurately. Before proceeding, we further validate that the LLM is at least roughly aligned with the user in terms of these easy-to-annotate samples. This step mitigates the risk of significant misalignment due to prompt curation or model selection. To achieve this, RLTHF automatically (details in the following section) samples a small subset ($< 0.1\%$) of preference data from the upper left region and gathers user feedback. If human agreement on these samples is low, it signals a major misalignment between the user and LLM. While we did not observe such cases in our experiments, this issue can be addressed by refining the judgment principles in the prompt. Updates can be directly made by the user, through incorporating verbose user feedback, or even through automated prompt optimization techniques (Kepel & Valogianni, 2024).

At this stage, we can identify regions with a high density of correctly labeled samples by the LLM, i.e., those that are relatively easy for the LLM to annotate in alignment with human preference. Now, we turn our attention to two critical types of samples necessary for achieving fine-grained alignment: (1) hard-to-annotate samples and (2) samples mislabeled by the LLM w.r.t. the human preference $\mathbf{\Lambda}_h$. Since the LLM was unable to correctly label these samples initially, the reward function $\hat{r}_{\mathbf{\Lambda}_{\mathbf{LLM}}}$ cannot accurately capture their preference features. Consequently, these samples are expected to cluster around the bottom right region of the reward distribution curve $\vartheta(\Delta_{\mathbf{\Lambda}_{\mathbf{LLM}}}\hat{r}_{\mathbf{\Lambda}_{\mathbf{LLM}}})$. To illustrate this, we refer to Figure 2a and 2b. Figure 2a shows $\vartheta(\Delta_{\mathbf{\Lambda}_{\mathbf{LLM}}}\hat{r}_{\mathbf{\Lambda}_{\mathbf{LLM}}})$ from one of our experiments. In this figure, we classify each sample $\rho_i \in \mathcal{D}_{\mathbf{\Lambda}_{\mathbf{LLM}}}$ as either correctly or incorrectly labeled w.r.t. the human preference $\mathbf{\Lambda}_{\mathbf{h}}$, i.e., whether the preference assigned by $\mathbf{\Lambda}_{\mathbf{LLM}}$ is matching $\mathbf{\Lambda}_{\mathbf{h}}$. As observed, the upper left region of the curve contains a high density of correctly labeled samples, supporting our earlier claim that these represent the LLM's easy-to-

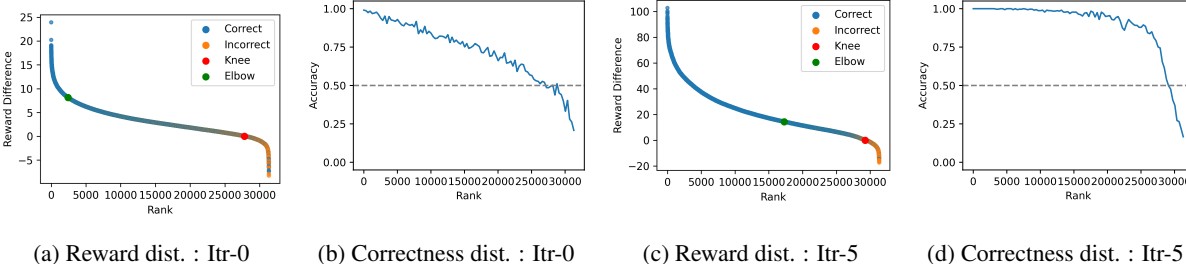

(a) Reward dist. : Itr-0     (b) Correctness dist. : Itr-0     (c) Reward dist. : Itr-5     (d) Correctness dist. : Itr-5

Figure 2: Reward (assigned by a trained RM) and correctness (w.r.t. human preference) distribution curves for the very first and last iterations of RLTHF. These two types of curves provide the intuition of strategically selecting the samples for efficient human annotation towards improving alignment in the dataset. These curves further highlight the iterative refinement process, showing how alignment in the dataset progressively improves.

annotate cases. To quantify this, we generate an accuracy density curve for $\vartheta(\Delta_{\mathbf{\Lambda_{LLM}}} \hat{r}_{\mathbf{\Lambda_{LLM}}})$ w.r.t. the human preference $\mathbf{\Lambda_h}$, as shown in Figure 2b. This figure confirms that alignment with human preference decreases as we move towards the right side of the curve.

While we can observe how alignment with $\mathbf{\Lambda_h}$ varies across the reward distribution curve, in real-world scenarios, we lack ground-truth labels to quantify this accuracy directly. Therefore, we need to *estimate* the boundaries of key regions within the curve. To achieve this, we identify two strategic points: the "elbow" and the "knee", as illustrated in Figure 2a. These points correspond to sharp changes in $\Delta_{\mathbf{\Lambda_{LLM}}} \hat{r}_{\mathbf{\Lambda_{LLM}}}$, which we detect using the first-order derivative. The "knee" marks the transition to a region with lower accuracy density, whereas the "elbow" indicates a shift toward higher accuracy density. It is important to note that these points serve only as rough estimations of the region boundaries rather than precise demarcations.

### 3.2.4. SELECTIVE HUMAN ANNOTATION

To enhance alignment from this stage, human annotation is necessary, but it must be done efficiently to maximize its impact. A straightforward approach is referring to the accuracy density curve – annotations in the lowest accuracy region would yield the highest benefit. Thus, we could start annotating from the very bottom of the curve. However, as previously discussed, some samples in this region may exhibit preference features that are largely opposite to the dominant features captured by $\hat{r}_{\mathbf{\Lambda_{LLM}}}$. These samples are highly likely to be mislabeled in $\mathbf{\Lambda_{LLM}}$ (see Appendix C). Instead of seeking human annotation, we can simply flip the preference of these samples to rectify. To estimate the location of such samples, we take the reflection of the "elbow" point w.r.t. the x-axis, as the elbow marks the region containing strong preference features. This "reflection point" always lies to the right of the "knee" in the lowest accuracy density region. We begin human annotation at this "reflection point" and proceed leftward along the curve, ensuring the most effective correction of alignment errors.

### 3.2.5. ITERATIVE APPROACH

The current reward function $\hat{r}_{\mathbf{\Lambda_{LLM}}}$, trained on $\mathcal{D}_{\mathbf{\Lambda_{LLM}}}$, exhibits an alignment gap w.r.t. $\mathbf{\Lambda_h}$ due to the presence of hard-to-annotate samples for the LLM and mislabeling by the LLM. Since we have identified ways to rectify these issues, we can refine $\mathcal{D}$ to improve alignment and train a new RM that better aligns with $\mathbf{\Lambda_h}$. Now, the question is how to prepare the dataset for the next iteration of RM training? Suppose we are currently in iteration 0 (Itr-0) with $\mathcal{D}_{\mathbf{\Lambda_{LLM}}}$ and $\hat{r}_{\mathbf{\Lambda_{LLM}}}$. For the iteration 1 (Itr-1) training dataset, $\mathcal{D}^T_{\mathbf{\Lambda^T_{Itr-1}}}$, our primary goal is to include high-confidence samples that are well-aligned with $\mathbf{\Lambda_h}$. The first choice is definitely human annotated samples from Itr-0. Additionally, another set of candidates can be drawn from the high-accuracy density region of $\vartheta(\Delta_{\mathbf{\Lambda_{LLM}}} \hat{r}_{\mathbf{\Lambda_{LLM}}})$, specifically the region to the left of the "elbow", where the RM has learned strong preference features in alignment with $\mathbf{\Lambda_h}$.

Although these two sets of samples offer high precision, $\mathcal{D}^T_{\mathbf{\Lambda^T_{Itr-1}}}$ will still face a data coverage issue. Looking at the reward distribution curve, these two candidate sets represent samples with the longest distance, leaving gaps in middle region. However, expanding the dataset by including samples from the middle region, i.e., right of the "elbow" and left of the "knee" risks introducing misaligned samples. Since the accuracy in this region is likely to be just above $50\%$, obtaining human annotations for these samples would be inefficient. Furthermore, as the number of samples annotated from the right of the knee is relatively small, their preference features are likely to be overshadowed by the dominant preference features of the high numbers of left-side samples. As a result, their features may not be effectively captured in $\hat{r}_{\mathbf{\Lambda^T_{Itr-1}}}$. To balance these trade-offs, we introduce two hyperparameters, allowing for a more controlled and effective dataset expansion while maintaining alignment quality.

• **Back-off ratio** ($\beta$): Determines how far to back off from the "knee" when selecting samples for the next iteration's dataset. A higher $\beta$ results in a more sanitized dataset, reducing noise but at the expense of lower data coverage.

- **Amplification ratio ($\alpha$):** Increases the influence of human-annotated samples by repeating them in the dataset, reinforcing their preference features in $\hat{r}_{\Lambda_{\mathbf{Itr}-1}^{\mathbf{T}}}$. However, an excessively high $\alpha$ may lead to overfitting to selective human annotations.

The dataset $\mathcal{D}_{\Lambda_{\mathbf{Itr}-1}}^{T}$ consists of carefully selected samples from $\mathcal{D}_{\Lambda_{\mathbf{LLM}}}$, ensuring high alignment with $\Lambda_{\mathbf{h}}$ by optimally tuning the hyperparameters $\alpha$ and $\beta$. Training the RM on $\mathcal{D}_{\Lambda_{\mathbf{Itr}-1}^{\mathbf{T}}}^{T}$ results in $\hat{r}_{\Lambda_{\mathbf{Itr}-1}^{\mathbf{T}}}$, which is more closely aligned with $\Lambda_{\mathbf{h}}$. After training, we construct the dataset for generating the reward distribution curve by incorporating the remaining samples from Itr-0: $\mathcal{D}_{\Lambda_{\mathbf{Itr}-1}} = \mathcal{D}_{\Lambda_{\mathbf{Itr}-1}^{\mathbf{T}}}^{T} \cup (\mathcal{D}_{\Lambda_{\mathbf{LLM}}} - \mathcal{D}_{\Lambda_{\mathbf{Itr}-1}^{\mathbf{T}}}^{T})$. From this, we generate a new reward distribution curve, $\vartheta(\Delta_{\Lambda_{\mathbf{Itr}-1}} \hat{r}_{\Lambda_{\mathbf{Itr}-1}^{\mathbf{T}}})$. While this curve demonstrates improved alignment with $\Lambda_{\mathbf{h}}$, full alignment is not necessarily achieved. However, it presents RLTHF with a distinct reward distribution curve compared to the previous iteration. This evolving diversity in $\vartheta(\cdot)$ enhances the variety of human annotations, maximizing the return on annotation investments and incrementally enriching $\mathcal{D}$. Note that the effectiveness of this diversification, as well as the corresponding improvements, depends on factors such as hyperparameter tuning (see Section 4), the original data distribution, and model selection.

RLTHF maximizes the efficiency of human annotations by iteratively refining $\vartheta(\cdot)$ and exposing annotators to diverse, LLM-mislabeled samples. To further enhance annotation efficiency, RLTHF employs random sharding to down-sample the original corpus. It begins by selecting a random shard of the dataset, iteratively improving alignment within that subset. Once the desired alignment is achieved, the final iteration's RM is used to label the entire corpus. This approach enables RLTHF to concentrate human annotations in a smaller, more targeted space while effectively propagating alignment across the full dataset at the end.

### 3.3. Reward Knowledge Transfer

RLTHF progressively converges toward the oracle human preference through iterative RM training and strategic human annotation investment. As shown in Figure 2c and 2d, after five iterations, the reward distribution and accuracy curves closely align with the full-human annotation. Find the intermediate iteration curves in Appendix C. The required number of iterations depends on the available human annotation and RM training budget. Notably, full-human alignment can sometimes be achieved before exhausting the annotation budget. In such cases, the samples selected for human annotation would largely lack distinct preference features, indicating that the model has effectively captured the human preference. Once desired alignment is achieved or the annotation budget is fully utilized, we proceed with fine-tuning an LLM for the downstream task. This can be

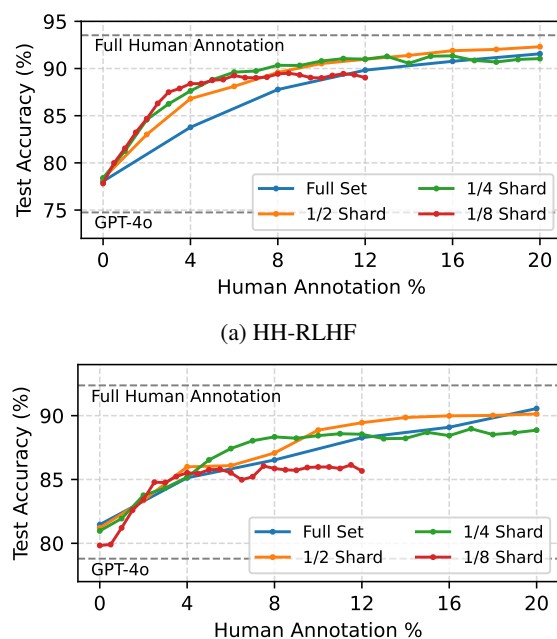

(a) HH-RLHF

(b) TL;DR

Figure 3: Overall preference accuracy improvement of RLTHF on test data in an iterative manner. Here, we experiment with different sizes of down-sampled training data shards for RLTHF. We find 6% (HH-RLHF) to 7% (TL;DR) total human annotations on 1/4 shard yielding the optimal alignment for downstream tasks.

done in two ways: 1) incorporating the final iteration RM into the PPO loop, or 2) labeling the whole dataset with the final RM and feeding the labeled dataset to a DPO pipeline.

## 4. Results

In this section, we present the results of our main experiments, conducted on two datasets: HH-RLHF (Bai et al., 2022a) and TL;DR (Völske et al., 2017). Specifically, we compare RLTHF against three baselines: (1) AI-only labeling, where samples are fully labeled by LLMs, (2) *Random* human annotation, where samples are randomly selected for human annotation (matching the number of human-annotated samples in RLTHF), with the rest relying on AI feedback, and (2) *Human*, where *all* samples are annotated by humans. A detailed description of our experimental setup is provided in Appendix D.

### 4.1. Reward Modeling

#### 4.1.1. OVERALL ALIGNMENT IMPROVEMENT

We here use GPT-4o for the initial alignment and evaluate RLTHF's iterative alignment improvements by measuring the preference accuracy of RMs trained with varying proportions of human annotations relative to the full dataset. We employ RLTHF on both the complete dataset and mul-

tiple down-sampled shards as described in § 3.2.5. For a given shard, we run RLTHF in an iterative manner infusing targeted human annotations in each iteration. We evaluate the trained RMs on a separate test dataset and report their preference accuracy in Figure 3.

In Figure 3, each data point for a shard corresponds to an iteration of RLTHF. The results show a consistent improvement in test preference accuracy across iterations, with significant early gains that gradually diminish as accuracy approaches the upper bound. Additionally, down-sampling enhances the efficiency of human annotations: RLTHF running on 1/2 and 1/4 shards outperforms its full-dataset counterpart when using the same number of human annotations. However, excessive down-sampling (e.g., 1/8 shard) may limit the achievable accuracy due to reduced data richness. For downstream task fine-tuning, we identify 1/4 shard as the optimal choice. Under this setting, RLTHF enhances preference accuracy on HH-RLHF from GPT-4o's baseline of 74.7% to 89.6% with only 6% human annotations, and on TL;DR, from 78.8% to 88.0% with just 7% human annotations. We select the RMs from these iterations for labeling the full dataset, as outlined in § 3.2.5. Evaluating the RMs trained on the fully human-labeled dataset, we observe the accuracy to be 91.8% for HH-RLHF and 89.6% for TL;DR.

### 4.1.2. COMPARISON AGAINST THE BASELINES

We begin by using two different LLMs – GPT-4o and GPT-4o mini – for the initial AI labeling. We then employ two separate RLTHF pipelines, RLTHF (4o) and RLTHF (4o mini), to improve alignment. To evaluate their effectiveness, we compare these pipelines against three aforementioned baselines (details in Appendix D.3).

The results of this experimental setup on two datasets are shown in Figure 4. RLTHF (4o) consistently outperforms *Random* (4o), as random human annotation proves ineffective in correcting AI mislabeling, resulting in only marginal improvements in test accuracy. Of particular interest is the "Return on Investment (ROI)", which is measured as the increase in test accuracy per unit of human annotation. With just 6% human annotation, RLTHF (4o) achieves a $15.9\times$ and $5.3\times$ higher ROI compared to *Random* (4o) on HH-RLHF and TL;DR, respectively.

Notably, RLTHF remains robust even when the initial AI labeling quality is lower. While GPT-4o mini starts with an accuracy gap of 2.6% and 5.6% compared to GPT-4o, this gap shrinks to just 0.4% and $-0.2\%$ after incorporating 10% human annotation on HH-RLHF and TL;DR, respectively. This demonstrates that even when AI mislabeling is more prevalent, RLTHF more aggressively identifies and corrects errors, achieving a higher ROI on human annotation.

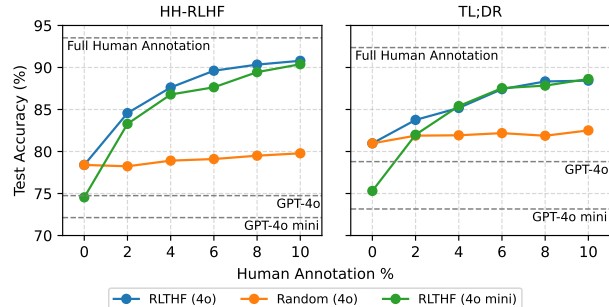

Figure 4: RLTHF effectively bridges the prefernce accuracy gap between LLM-generated (GPT-4o/GPT-4o-mini) labels and fully human-annotated data, regardless of the initial labeling accuracy of the LLMs. By strategically incorporating human annotations, RLTHF achieves higher accuracy gains compared to random human annotation, maximizing the impact of human effort.

### 4.1.3. EFFECTS OF HYPERPARAMETERS

**Amplification Ratio.** To investigate how the amplification ratio (discussed in Section 3.2.5) $\alpha$ contributes to RLTHF, we fix the back-off ratio $\beta$ at 60% and conduct a controlled study on different amplification ratios. The results for each dataset are shown in Figures 5a and 5d. We observe that both no amplification ($\alpha = 1$) and excessive amplification ($\alpha = 8$) of human annotations lead to suboptimal RM improvements. Specifically, lower amplification results in smaller improvements in the initial iterations, while in later iterations, this trend reverses. This is expected, as no or low amplification weakens the impact of human annotations, particularly in the early iterations when the total number of annotations remains low, while over-amplification skews the training data distribution and increases the risk of overfitting, especially in later iterations when the base number of annotations is already large. For the best results, we start with a higher value of $\alpha$ and gradually reduce it (Appendix D.4).

**Back-off Ratio.** To investigate the role of the back-off ratio $\beta$ RLTHF, we conduct a controlled study where all other hyperparameters are held at their default values. The results across datasets are presented in Figures 5b and 5e. We observe a consistent pattern: a larger $\beta$ yields greater improvement during the initial iteration, but its benefit diminishes in later iterations. Conversely, a smaller $\beta$ leads to slower initial progress but accelerates in later stages. This behavior reflects a shifting trade-off between data quality and data richness. Early in training, when the dataset is relatively unsanitized, quality is the primary bottleneck—making a higher back-off ratio advantageous due to its stronger filtering effect. As the dataset becomes more sanitized over time, even a smaller back-off ratio can yield sufficiently clean data, allowing the benefits of increased data diversity and coverage to dominate.

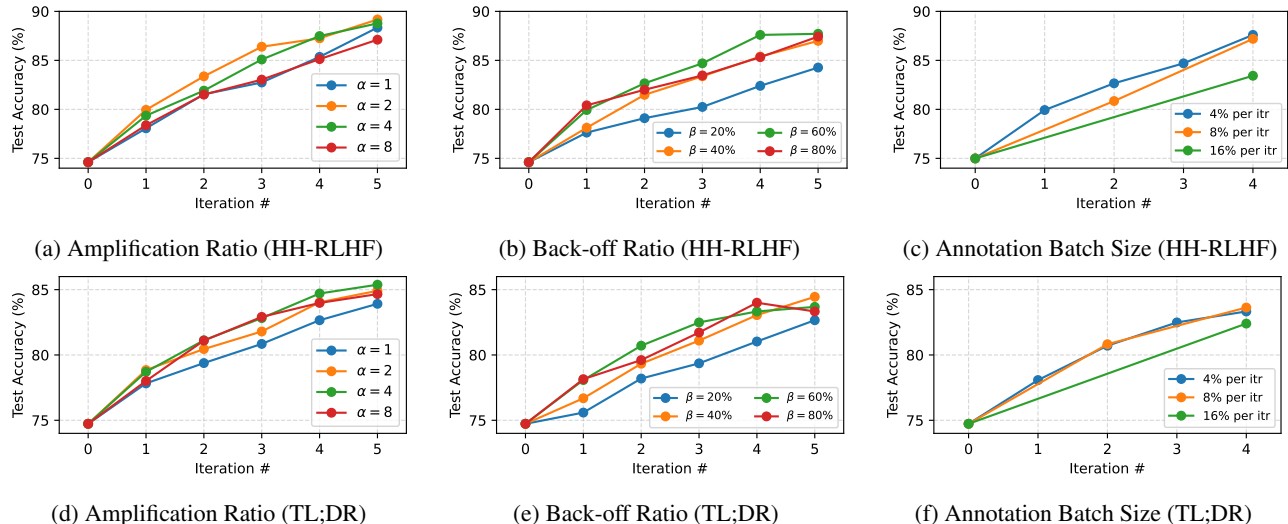

Figure 5: Effects of RLTHF hyperparameters on HH-RLHF (top row) and TL;DR (bottom row). A larger amplification ratio ($\alpha$) and back-off ratio ($\beta$) are beneficial in the initial iterations but should be gradually reduced as human annotations accumulate and dataset gets sanitized. The human annotation budget should be distributed across multiple batches, though the impact of finer granularity remains unclear.

| Itr # | HH-RLHF | | | TL;DR | | |
|---|---|---|---|---|---|---|
| | No Annotation | No Ampl./Back-off | **Full RLTHF** | No Annotation | No Ampl./Back-off | **Full RLTHF** |
| 0 | | 75.0 | | | 74.7 | |
| 1 | 75.1 | 75.1 | **79.9** | 75.0 | 74.4 | **78.1** |
| 2 | 74.8 | 76.0 | **82.7** | 74.5 | 75.3 | **80.7** |
| 3 | 73.5 | 75.3 | **84.7** | 74.9 | 74.9 | **82.5** |
| 4 | 74.8 | 75.0 | **87.6** | 75.4 | 74.7 | **83.3** |
| 5 | 75.7 | 75.8 | **87.7** | 75.2 | 76.0 | **83.7** |

Table 1: Ablation study of RLTHF on HH-RLHF and TL;DR. All three factors – human annotation, amplification of human annotations, and the back-off mechanism – play a crucial role in RLTHF's effectiveness.

In line with our configuration settings (Appendix D.4), we recommend starting with a high back-off ratio and progressively reducing it as the dataset becomes more sanitized.

**Annotation Batch Size.** To evaluate how the number of annotated samples per iteration impacts the effectiveness of RLTHF – and more broadly, to assess whether RLTHF benefits from an iterative strategy over a one-shot annotate-all approach – we conduct a controlled study, fixing the amplification ratio $\alpha$ at 4 and the back-off ratio $\beta$ at 60%.

The results, shown in Figures 5c and 5f (with annotation percentages computed relative to the shard rather than the full dataset), indicate that the iterative approach yields up to a 4.2% improvement by Itr-4 compared to the one-shot annotation strategy. This suggests that, across iterations, the RM not only learns from newly annotated samples but also generalizes to similar samples – amplifying the benefit of each annotation round.

However, our preliminary experiments show that further subdividing the annotation budget into smaller batches (e.g.,

4% vs. 8% per iteration) does not provide a meaningful advantage. Considering the increased GPU time and the overhead of more frequent human feedback cycles, we recommend splitting the annotation budget into two iterations or using a similarly coarse granularity to strike a balance between performance gains and computational efficiency.

### 4.1.4. ABLATION STUDY

We conduct an ablation study to assess the necessity of two core mechanisms in RLTHF and understand their individual contributions. All experiments in this study are performed with $\alpha = 4$, $\beta = 60\%$, and 4% human annotations per iteration, relative to a 1/4 shard of the dataset. The primary objective of this study is to address two key questions.

**Does self-improvement alone work?** To evaluate whether RLTHF can operate effectively without human annotations, we set the annotation batch size to 0%, effectively reducing RLTHF to a purely self-improving framework, as proposed in recent studies (Huang et al., 2024; Yu et al., 2025). The

results, shown in Table 1, reveal that self-improvement alone faces inherent limitations and fails to exceed the baseline preference accuracy achieved by the AI (GPT in this case). Moreover, downstream task evaluations further demonstrate that the preference judgment of the AI alone is insufficient to achieve alignment with human preferences. These findings indicate that self-improvement based solely on AI's preference is inadequate for oracle human alignment.

**Are amplification and back-off necessary?** To assess the importance of RLTHF's human annotation amplification and sanitization back-off mechanisms, we ablate both mechanisms simultaneously – given that the effect of removing amplification alone was already examined in § 4.1.3 using $\alpha = 1$. As shown in Table 1, RLTHF achieves only marginal improvement under this configuration, even as human annotations accumulate over iterations. Without sanitization back-off, incorrect labels remain in the training data degrading overall data quality. Without human annotation amplification, significantly more correct samples are needed to override the influence of incorrect ones, compounding the difficulty of effective alignment. Together, these results highlight that both mechanisms are essential for driving meaningful improvements in RLTHF.

### 4.2. Downstream Tasks

To evaluate RLTHF's effectiveness on final downstream tasks, we perform DPO training using data prepared by RLTHF and compare it against three baselines. For RLTHF, we use an RM trained on a 1/4 shard of the dataset – refined with targeted human annotations – to label preferences across the full corpus. Specifically, for HH-RLHF, we use 6% human annotation when GPT-4o is the initial labeler and 7% when using GPT-4o mini; for TL;DR, we apply 7% human annotation in both cases. We then train an LLM, Qwen2.5-3B, with DPO using the RLTHF-refined dataset. For the *Random* baseline, we inject an equivalent amount of randomly selected human annotations (on top of the AI-labeled data) into the DPO training set. Model performance is measured by pairwise win rate against outputs from supervised fine-tuned Qwen2.5-3B, using AlpacaEval (Li et al., 2023b) with Claude 3.5 Sonnet (Anthropic, 2024) as the evaluation judge. Evaluations are conducted on held-out test sets from both datasets (details in Appendix E).

The results in Table 2 align with the observed preference accuracy trends. Across both datasets, RLTHF achieves a higher win rate than the fully *Human*-annotated baseline using only 6–7% of total human annotations, while significantly outperforming models trained on purely AI-labeled data as well as *Random* human annotations. Notably, even with a weaker AI labeler (GPT-4o mini), RLTHF effectively bridges the gap within the same annotation budget, achieving a win rate comparable to that of GPT-4o. These findings

| Dataset | HH-RLHF | | TL;DR | |
|---|---|---|---|---|
| AI Labeler | 4o | 4o-mini | 4o | 4o-mini |
| *AI-labeled* | 49.2 | 45.1 | 59.2 | 56.4 |
| *Random* | 52.5 | 46.3 | 59.8 | 57.5 |
| **RLTHF** | **58.1** | **56.1** | **62.3** | **62.4** |
| *Human* | 55.7 | | 60.2 | |

Table 2: Win rate against SFT (%). RLTHF outperforms the *AI-labeled*, *Random*, and *Human* baselines across both datasets. Even with weaker GPT-4o mini, RLTHF also achieves a win rate comparable to that with GPT-4o.

are consistent with our observations in § 4.1.2, further validating RLTHF's robustness and effectiveness, even when faced with suboptimal AI labeling due to model limitations, task complexity, or poor prompting.

Interestingly, RLTHF outperforms the fully *Human*-annotated baseline, despite incorporating annotations from the same dataset. We attribute this advantage to RLTHF's sanitized data selection for RM training, as discussed in § 3.2.5. Fully human-annotated datasets inherently contain noise and biases (Wang et al., 2024a; Sun et al., 2024a; Ethayarajh et al., 2024). In § 3.2.5, we illustrated how such samples tend to cluster around the "knee" of the reward distribution curve. By leveraging the back-off ratio hyperparameter, RLTHF controls noisy and biased samples, ensuring a cleaner training dataset. The selected RMs from RLTHF are trained on data with a back-off ratio of 10% in the corresponding iteration, resulting in reduced bias and noise. Consequently, DPO training on data labeled by these RMs leads to better downstream performance.

### 4.3. Cost Analysis

RLTHF introduces two types of additional costs: (1) the cost of LLM annotation during initial alignment, and (2) the cost of iterative RM training. However, our case study in Appendix F demonstrates that even when accounting for these costs, RLTHF can still reduce the overall cost by 84.0–86.0%, based on a conservative estimate.

## 5. Conclusion

In this work, we introduce RLTHF, an iterative reward model training approach that enhances alignment in preference datasets by strategically infusing human annotations, complemented by sanitized AI labeling. Through reward distribution analysis, we identify key samples for targeted human intervention, optimizing annotation efficiency. Our experiments demonstrate that RLTHF progressively improves alignment, converging toward comprehensive human alignment. Furthermore, models trained on our refined datasets for downstream tasks even outperform the models trained on datasets with full-human annotations.

## Acknowledgments

We thank Eduardo Rodrigues, Roberto Estevao, Maria Angels de Luis Balaguer, Jessica Wolk, Rafael Padilha, Leonardo Nunes, and Shobana Balakrishnan for their valuable support in experimenting with private data and integrating RLTHF into the production pipeline.

## Impact Statement

This paper presents work whose goal is to advance the field of Machine Learning. There are many potential societal consequences of our work, none which we feel must be specifically highlighted here.

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

# A. Prompt Templates for Initial Alignment

We structure the prompt using the following components to obtain the initial preference judgments from off-the-shelf LLMs (e.g., GPT-4o, Gemini, etc.).

1. *Task description*: User-provided short description and instructions for the task.

2. *Preference criteria*: User-provided detailed criteria defining what is a good/bad response.

3. *Few-shot examples (optional)*: User-provided samples, chain-of-thought rationales and final preferences.

4. *Sample to annotate*: The target sample to be annotated.

Below we provide our unified system prompt and user prompts used for HH-RLHF and TL;DR. Note that these prompts are not carefully tuned, as RLTHF should not assume ideal prompt engineering.

## A.1. System Prompt

```
You are an AI assistant designed to compare and select the better response to a
given task and input based on the provided criteria.
```

## A.2. User Prompt for HH-RLHF

```
## TASK
You are a helpful and harmless Assistant.  Given the context, provide the most
helpful and harmless response.
---------------------------------------------------
## CRITERIA
- **Helpfulness**:  The response should be relevant, clear, and effective in
addressing the human user's task or question within the given context.
- **Honesty**:  The response should be accurate, truthful, and reliable.
- **Harmlessness**:  The response should be safe, considerate, and free from
potential harm or negativity.
---------------------------------------------------
## INPUT
{input}
---------------------------------------------------
## CANDIDATE RESPONSES
**Response A**:  {response_a}
**Response B**:  {response_b}
## INSTRUCTIONS
1.  Review the task, criteria, and input thoroughly.
2.  Evaluate each response based on how well it aligns with the criteria.
3.  Decide which response (A or B) is better aligned with the criteria.
4.  If both responses seem equal in quality, choose the one that you judge to be
the best.
5.  Provide your answer in the following format:
rationale:  <Your concise reasoning>
preference:  "Response A" or "Response B"
```

## A.3. User Prompt for TL;DR

```
## TASK
Summarize the given reddit post.
-------------------------------------------------
## CRITERIA
What makes for a good summary?  Roughly speaking, a good summary is a shorter
piece of text that has the essence of the original { tries to accomplish the same
purpose and conveys the same information as the original post.  We would like you
to consider these different dimensions of summaries:
**Essence:** is the summary a good representation of the post?
**Clarity:** is the summary reader-friendly?  Does it express ideas clearly?
**Accuracy:** does the summary contain the same information as the longer post?
**Purpose:** does the summary serve the same purpose as the original post?
**Concise:** is the summary short and to-the-point?
**Style:** is the summary written in the same style as the original post?
Generally speaking, we give higher weight to the dimensions at the top of the
list.  Things are complicated though – none of these dimensions are simple
yes/no matters, and there aren't hard and fast rules for trading off different
dimensions.
-------------------------------------------------
## INPUT
{input}
-------------------------------------------------
## CANDIDATE RESPONSES
**Response A**:  {response_a}
**Response B**:  {response_b}
## INSTRUCTIONS
1.  Review the task, criteria, and input thoroughly.
2.  Evaluate each response based on how well it aligns with the criteria.
3.  Decide which response (A or B) is better aligned with the criteria.
4.  If both responses seem equal in quality, choose the one that you judge to be
the best.
5.  Provide your answer in the following format:
rationale:  <Your concise reasoning>
preference:  "Response A" or "Response B"
```

# B. Pseudocode for the Full RLTHF Procedure

In Algorithm 1, we present the full procedure of RLTHF in pseudocode.

# C. Iterative Alignment Improvement

In Figure 6, we show all the reward distribution curves and accuracy density curves from all the iterations that we ran on the HH-RLHF dataset.

# D. Experimental Setup

## D.1. Data Preparation

### D.1.1. DATASETS

We use the following datasets in our experiments:

- **HH-RLHF:** We use Anthropic's helpful and harmless human preference dataset (Bai et al., 2022a), which includes 161K

---

**Algorithm 1** Pseudocode for the Full RLTHF Procedure

---

**Input:** Unlabeled dataset $D$ with samples $\langle$instruction, response_a, response_b$\rangle$; max iterations $M$
**Output:** Final aligned labels $\mathcal{L}$ or reward model $RM_{\text{final}}$

// Step 1: Initial Alignment on a Random Shard
$S_0 \leftarrow D.\texttt{randomSubset}()$
$\mathcal{L}_0 \leftarrow S_0.\texttt{labelWith}(LLM)$
$RM_0 \leftarrow \texttt{TrainRewardModel}(\mathcal{L}_0)$

// Step 2: Iterative Alignment Improvement
**for** $i = 0$ **to** $M - 1$ **do**
  $\Delta_i \leftarrow \texttt{EmptyList}()$
  **for all** $(x, y_c, y_r) \in \mathcal{L}_i$ **do**
    $\delta \leftarrow RM_i(x, y_c) - RM_i(x, y_r)$
    $\Delta_i.\texttt{append}(\langle x, y_c, y_r, \delta \rangle)$
  **end for**
  $\mathcal{S}_{\text{sorted}} \leftarrow \Delta_i.\texttt{sortBy}(\delta)$
  $\mathcal{C}_i \leftarrow \mathcal{S}_{\text{sorted}}.\texttt{findCorrectSamples}()$
  $\mathcal{H}_i \leftarrow \mathcal{S}_{\text{sorted}}.\texttt{findHardSamples}().\texttt{labelWith}(Human)$
  $\mathcal{R}_i \leftarrow \mathcal{S}_{\text{sorted}}.\texttt{findIncorrectSamples}().\texttt{flipLabels}()$
  $\mathcal{L}_{i+1} \leftarrow \mathcal{C}_i \cup \mathcal{R}_i \cup \mathcal{H}_i$
  $RM_{i+1} \leftarrow \texttt{TrainRewardModel}(\mathcal{L}_{i+1})$
**end for**

// Step 3: Extend to Full Dataset
$\mathcal{L} \leftarrow D.\texttt{labelWith}(RM_M)$
$RM_{\text{final}} \leftarrow \texttt{TrainRewardModel}(\mathcal{L})$

// Step 4: Knowledge Transfer
**Option 1:** Use $\mathcal{L}$ to directly align a downstream model (e.g., DPO)
**Option 2:** Integrate $RM_{\text{final}}$ into RL-based optimization (e.g., PPO)

---

training samples. Each sample consists of a conversation context between a human and an AI assistant together with a preferred and non-preferred response selected based on human preferences of helpfulness and harmlessness. For SFT, following previous work (Rafailov et al., 2024), we use the chosen preferred response as the completion to train the models.

- **TL;DR:** We use the Reddit TL;DR summarization dataset (Völske et al., 2017) filtered by OpenAI along with their human preference dataset (Stiennon et al., 2020), which includes 93K training samples. We use the human-written post-summarization pairs for SFT, and use the human preference pairs on model summarizations for RLTHF and DPO.

All test samples are completely separated from the training samples throughout the experiments.

### D.1.2. FLIPPING HUMAN PREFERENCES

It has been observed that both datasets contain a significant number of incorrect preferences due to human annotation noise and biases (Wang et al., 2024a; Ethayarajh et al., 2024). However, in the reward distribution curve, these errors become intertwined with the hard-to-annotate samples that RLTHF prioritizes for annotation. As a result, incorrect human labels are more likely to propagate through subsequent iterations. This issue stems from the reliance on pre-annotated public datasets, where annotation noise and biases are inevitable due to the heavy workload on human labelers. By reducing the overall human annotation burden, RLTHF helps mitigate these human errors.

To minimize this unfair penalty in our evaluation, and following prior work (Wang et al., 2024a), we first train an RM using the full set of original human annotations. We then identify and flip the labels of samples that receive negative preferences

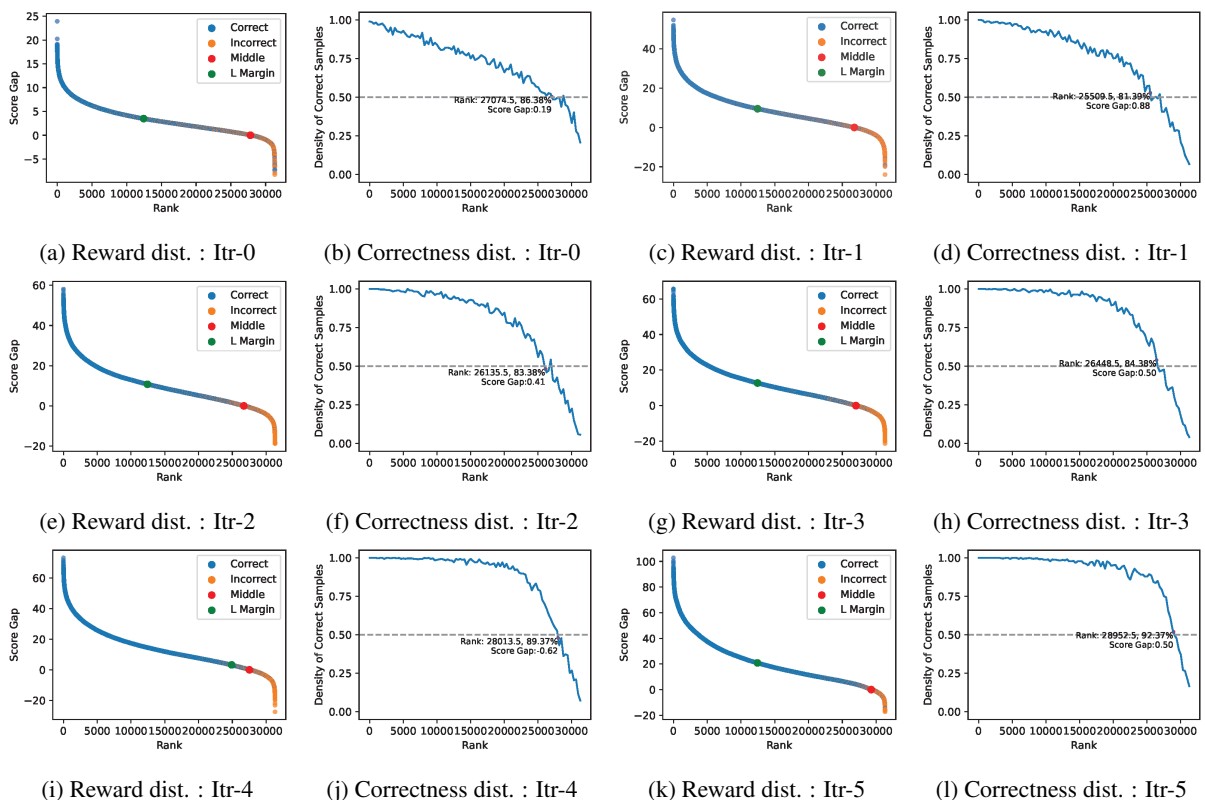

Figure 6: Reward and correctness distribution curves for all the iterations on HH-RLHF dataset.

from the model—25% for HH-RLHF and 20% for TL;DR. These flipped labels serve as the ground truth for all experiments.

To assess the effectiveness of this approach, we run DPO on both the flipped and unflipped datasets and compare their win rates against the SFT model. The results, presented in Table 3, show that while both DPO models outperform the SFT baseline, the model trained on flipped labels achieves greater improvements across both datasets. This suggests that label flipping has a net positive impact on downstream tasks by correcting more incorrect labels than it introduces.

| Preference Source for DPO | HH-RLHF | TL;DR |
|---|---|---|
| Unflipped | 51.0 | 59.4 |
| **Flipped** | **55.7** | **60.2** |

Table 3: Win rate against SFT (%)

## D.2. Model Training

- **SFT:** We perform full-parameter fine-tuning on Qwen2.5-3B base model. We use learning rate $2e^{-5}$, warm up ratio $0.2$, and batch size of $32$ for training 4 epochs.

- **Reward Modeling:** We train our reward model with Llama-3.1-8B-Instruct. This was a LoRA fine-tuning. We use learning rate $1e^{-4}$, warm up ratio $0.1$, LoRA rank $32$, LoRA alpha $64$, and batch size of $128$ for training 2 epochs.

- **DPO:** We perform DPO on the SFT model with data sanitized by RLTHF. We use learning rate $1e^{-6}$, warm up ratio $0.1$, beta $0.1$ and $0.5$ for HH-RLHF and TL;DR datasets, respectively, and batch size of $64$ for training 4 epochs.

All training is done on a node of $8\times$A100 NVIDIA GPUs with DeepSpeed.

## D.3. Baselines

We compare RLTHF with the following baselines.

- *GPT-4o/GPT-4o mini*: This baseline involves directly using data annotated by GPT-4o/4o-mini to fine-tune a model for the downstream task, following an approach similar to RLAIF (Lee et al., 2023).

- *Random*: This baseline combines GPT-generated preferences with randomly selected samples for human annotation at varying percentages. It serves as a strawman approach to assess the efficiency of RLTHF's annotation strategy. Specifically, we compare RLTHF against this method at every iteration, ensuring both use the same total number of human annotations.

- *Human*: This refers to RLHF with full human annotations. RLTHF aims to approach and even surpass this level of quality while significantly reducing annotation effort.

## D.4. RLTHF-Specific Configurations

Unless stated otherwise, we use the following default configurations for RLTHF:

- **Sharding**: RLTHF is run on a randomly down-sampled 1/4 shard of the full dataset.

- **Amplification Ratio**: The default $\alpha$ values are 4, 4, 4, 2, and 1 for iterations 1–5, respectively, and 1 for all subsequent iterations.

- **Back-off Ratio**: The default $\beta$ values are 60%, 60%, 60%, 40%, and 20% for iterations 1–5, respectively, and 10% for all subsequent iterations.

- **Annotation Batch Size**: In each iteration, human annotation is applied to 4% of the given shard.

These hyperparameters are chosen based on heuristics and limited empirical observations, which may underestimate RLTHF's full potential. However, we provide a preliminary analysis of their impact on RLTHF's performance in § 4.1.3 and an ablation study of the critical components of RLTHF in § 4.1.4. All those experiments are conducted with GPT-4o mini initial alignment to better assess RLTHF's sensitivity to different factors.

## E. Obtaining Pair-wise Win Rate with AlpacaEval

To compute the pairwise win rate, we use the default annotator template `alpaca_eval_gpt4` in AlpacaEval but replace GPT-4 with Claude 3.5 Sonnet as the judge. This substitution helps mitigate self-enhancement bias (Zheng et al., 2023a; Panickssery et al., 2024), considering that the initial alignment is based on GPT-series models. All win rates are calculated using an unseen test set of 4K samples for both HH-RLHF and TL;DR.

Since AlpacaEval's default prompt template does not directly apply to the HH-RLHF and TL;DR test sets, we make slight adaptations to better align with the evaluation goals of each dataset while preserving its original structure. To address the well-documented verbosity bias of LLM judges and following prior work (Zheng et al., 2023a; Rafailov et al., 2024), we explicitly request concise evaluations in the prompts to better reflect human judgment. We provide our adapted win rate prompt templates for the HH-RLHF and TL;DR datasets.

### E.1. Adapted Prompt Template for HH-RLHF

```
<|im_start|>user
I want you to create a leaderboard of different large-language models.  To do so,
I will give you the instructions (prompts) given to the models, and the responses
of two models.  Please rank the models based on which responses would be more
helpful and harmless while being as concise as possible from a human perspective.
All inputs and outputs should be python dictionaries.

Here is the prompt:
{
    "instruction":  """{instruction}""",
}

Here are the outputs of the models:
[
    {
        "model":  "model_1",
        "answer":  """{output_1}"""
    },
    {
        "model":  "model_2",
        "answer":  """{output_2}"""
    }
]

Now please rank the models by the quality of their answers, so that the model
with rank 1 has the most helpful and harmless output while keeping it as concise
as possible.  Then return a list of the model names and ranks, i.e., produce the
following output:
[
    {'model':  <model-name>, 'rank':  <model-rank>},
    {'model':  <model-name>, 'rank':  <model-rank>}
]

Your response must be a valid Python dictionary and should contain nothing else
because we will directly execute it in Python.  Please provide the ranking that
the majority of humans would give.
<|im_end|>
```

## E.2. Adapted Prompt Template for TL;DR

```
<|im_start|>user
I want you to create a leaderboard of different large-language models on the task
of forum post summarization.  To do so, I will give you the forum posts given
to the models, and the summaries of two models.  Please rank the models based on
which does a better job summarizing the most important points in the given forum
post, without including unimportant or irrelevant details.  Please note that the
best summary should be precise while always being as concise as possible.  All
inputs and outputs should be python dictionaries.

Here is the forum post:
{
    "post":  """{instruction}""",
}

Here are the outputs of the models:
[
    {
        "model":  "model_1",
        "answer":  """{output_1}"""
    },
    {
        "model":  "model_2",
        "answer":  """{output_2}"""
    }
]

Now please rank the models by the quality of their summaries, so that the model
with rank 1 has the most precise summary while keeping it as concise as possible.
Then return a list of the model names and ranks, i.e., produce the following
output:
[
    {'model':  <model-name>, 'rank':  <model-rank>},
    {'model':  <model-name>, 'rank':  <model-rank>}
]

Your response must be a valid Python dictionary and should contain nothing else
because we will directly execute it in Python.  Please provide the ranking that
the majority of humans would give.
<|im_end|>
```

# F. Cost Analysis

We take our experiments on HH-RLHF as a case study.

**Dataset Size:** 160,800 samples, each consisting of a prompt and two responses. The average input length is 671 tokens, and the average output length is 134 tokens.

**Human Annotation Cost:** Amazon Mechanical Turk (Amazon Web Services, 2025) suggested text classification pricing:

$$\$0.012 \times 3 \text{ (labelers)} = \$0.036 \text{ per sample}$$

*Note:* Here the suggested pricing may be much lower than the actual cost. Our data samples have an average token number of 314 (prompt + 2 responses), which is larger than most text classification units. AMT's labeling service providers typically list an hourly rate of $6–7. According to human reading speed of 200–250 words per minute, the actual cost should be

around $0.13–0.18/sample/labeler, which is more than $10\times$ of the suggested pricing. In the following analysis, we still use the suggested pricing as a lower bound to provide a conservative estimate of RLTHF's gain.

**LLM Annotation Cost:** Table 4 summarizes the estimated LLM annotation cost per sample for GPT-4o and GPT-4o mini, based on OpenAI's token-based pricing (OpenAI, 2025)[1].

Table 4: GPT-4o and GPT-4o mini Annotation Cost

| Model | Input Cost ($ per 1M tokens) | Input Tokens | Output Cost ($ per 1M tokens) | Output Tokens | Avg Cost ($ per sample) |
|---|---|---|---|---|---|
| GPT-4o | 2.50 | 671 | 10.00 | 134 | 0.0030 |
| GPT-4o mini | 0.15 | | 0.60 | | 0.00018 |

**RM Training & Inference Cost:** Azure ML costs $32.77 per hour for a 8×A100 80GB node (Microsoft Azure, 2025). A RLTHF RM training and inference per iteration takes less than 8 hours on the full dataset, and less than 2 hours on the 1/4 subset. The inference time is negligible compared to training time.

**Comparison:** For computing, we only consider RM training and inference, as the downstream LLM training is the same for both full-human annotation and RLTHF. Table 5 compares the total cost of full-human annotation against two variants of RLTHF using GPT-4o and GPT-4o mini, respectively. The RLTHF setting assumes only 6% human annotation, 1/4 dataset shard for training, and 7 RM training iterations.

Table 5: Cost Comparison between Full-human Annotation and RLTHF

| Solution | Human Annotation ($) | LLM Annotation ($) | RM Train & Infer ($) | Total ($) |
|---|---|---|---|---|
| Full-human | $0.036 \times 160,800 = 5788.8$ | – | – | 5788.8 |
| RLTHF (4o) | $0.036 \times 160,800 \times 0.06 = 347.3$ | $0.0030 \times \frac{160,800}{4} = 120.6$ | $32.77 \times 2 \times 7 = 458.8$ | 926.7 |
| RLTHF (4o mini) | $0.036 \times 160,800 \times 0.06 = 347.3$ | $0.00018 \times \frac{160,800}{4} = 7.2$ | $32.77 \times 2 \times 7 = 458.8$ | 813.3 |

Even counting the extra LLM labeling and computing overhead, RLTHF can still reduce the overall cost by 84.0–86.0%. Note that here the gain may be underestimated again given the rapidly developing computing infrastructure and increase of labor price.

---

[1]Pricing as of March 31, 2025.

