# OpenReview forum: "RLTHF: Targeted Human Feedback for LLM Alignment"
_ICML.cc/2025/Conference — ICML 2025 poster_

### Official Review · Reviewer_qWiJ · 2025-03-14

**Overall Recommendation:** 4

**Summary:**

This paper presents Sargy, a hybrid framework designed to align LLMs with human preferences by integrating LLM-generated annotations and selective human feedback. The framework operates iteratively, identifying erroneous samples through reward model distributions, prioritizing difficult cases for human annotation, and retaining accurate LLM labels. Experiments on HH-RLHF and TL;DR datasets show that Sargy achieves Oracle-level alignment quality with only 15-20% human annotations, while downstream models trained on Sargy-curated data perform comparably to fully human-annotated benchmarks.

**Claims And Evidence:**

The claims are well-supported by clear and convincing evidence.

**Essential References Not Discussed:**

None.

**Experimental Designs Or Analyses:**

1. Reward Model Iterative Improvement Experiment: Evaluated Sargy's iterative improvements on HH-RLHF and TL;DR datasets, showing it achieves near-oracle accuracy with only 20% human annotations, significantly outperforming random sampling. Experimental design is sound.

2. Amplification Ratio Experiment: Investigated the impact of different amplification ratios on reward model improvement, finding that using a higher ratio initially and reducing it later maximizes annotation effectiveness and model performance. Experimental design is sound.

3. Back-off Ratio Experiment: Analyzed the effect of different back-off ratios on data sanitization and model improvement, demonstrating that a higher ratio initially and reducing it later optimizes data quality and model performance. Experimental design is sound.

4. Annotation Batch Size Experiment: Explored the impact of batch size per iteration on model improvement, validating that iterative annotation outperforms one-shot annotation, enhancing model efficiency. Experimental design is sound.

5. Ablation Study: Verified the necessity of Sargy's components, including selective human annotation, amplification ratio, and back-off ratio, proving these are critical to Sargy's success. Experimental design is sound.

6. Downstream Task Experiment: Used Sargy's curated preference dataset for Direct Preference Optimization (DPO), evaluating model performance on HH-RLHF and TL;DR downstream tasks, showing results close to the oracle model and significantly better than random sampling and initial models. Experimental design is sound.

Overall, I find the experimental section well-designed and convincing, effectively supporting the paper's conclusions.

**Methods And Evaluation Criteria:**

Yes. This paper alleviates the challenge of fine-tuning LLMs to match user preferences, which is hindered by the high cost of high-quality human annotations in reinforcement learning human feedback (RLHF) and the limited generality of AI feedback. To overcome these issues, it propose a human-machine hybrid framework that leverages LLM-based initial alignment combined with selective human annotation to achieve near-human annotation quality with minimal effort.

**Other Comments Or Suggestions:**

None.

**Other Strengths And Weaknesses:**

None.

**Questions For Authors:**

None.

**Relation To Broader Scientific Literature:**

1. Relation to RLHF and RLAIF: The proposed Sargy framework integrates the strengths of RLHF (Reinforcement Learning from Human Feedback) and RLAIF (Reinforcement Learning from AI Feedback), addressing the high cost of human annotations in RLHF and the limited generalizability of AI feedback in RLAIF. By introducing a human-AI hybrid annotation strategy, Sargy achieves near-human annotation quality with minimal human effort, aligning with ongoing research on effectively combining human and AI feedback.

2. Relation to LLM Self-Improvement Methods: Sargy enhances LLM performance through iterative reward model training and selective human annotations. This approach resonates with recent advancements in LLM self-improvement (e.g., Self-Rewarding LMs and SELF-ALIGN) but distinguishes itself by incorporating human intelligence, overcoming the inherent limitations of LLM self-improvement, particularly in customized tasks, thereby advancing the field of LLM self-enhancement.

**Theoretical Claims:**

The paper presents a methodological framework (Sargy). The claims made in the paper are primarily empirical, based on experimental results.

In these claims, the rationale for segmenting the reward distribution curve could benefit from greater rigor, as the identification of "elbow" and "knee" points in the curve is heuristic and may not always correspond to clear boundaries between correctly and incorrectly labeled samples.

The remaining claims (Sarge's iterative alignment improvement effectiveness and knowledge transfer effectiveness) are all supported by experimental data.

---

> ### Author Rebuttal · Authors · 2025-04-01
>
> Thank you for your recognition and constructive review of our work!
>
> **Q1:** The identification of "elbow" and "knee" points in the curve is heuristic and may not always correspond to clear boundaries between correctly and incorrectly labeled samples.
>
> **Re:** We acknowledge the practical concern that detecting "elbow" and "knee" points may not be highly precise. Therefore, in our implementation, Sargy treats these points as approximate boundary estimates, i.e., we don't need a precise boundary. Empirically, we observed that "elbow" and "knee" yield satisfactory results and slight adjustments to these estimations do not affect performance. We'll add the corresponding numbers in the final version.

---

### Official Review · Reviewer_BWsT · 2025-03-15

**Overall Recommendation:** 4

**Summary:**

This paper proposes Sargy, a human-AI hybrid framework that combines LLM-based initial alignment with selective human annotations to achieve near-human annotation quality with minimal effort. The reward model's distribution is used to identify hard-to-annotate samples mislabeled. Then it iteratively enhances data quality by integrating strategic human corrections while leveraging LLM's correctly labeled samples.


## update after rebuttal: Thank you for your answer. I keep my score as is.

**Claims And Evidence:**

1. Using the reward model's distribution to select hard-to-annotate samples
- Yes
2. An iterative reward model training technique to achieve oracle-level human alignment in the dataset.
- Yes
3. Sargy is implemented on HH-RLHF and TL;DR. Results show accuracy comparable to fully human-annotated oracle dataset while using 20% of the total human annotations.
- Yes

**Essential References Not Discussed:**

No

**Experimental Designs Or Analyses:**

The proposed method is applied on HH-RLHF and TL;DR. Baselines are Random, and Oracle. I would appreciate if the authors include other variants that are e.g., Greedy where the upper-left or bottom-right samples of the reward distribution are being annotated. I would also encourage the authors to report performance for more than 5 iterations. Obtained performance are better than Random, which is not a surprise. Finally, the ablation study is sound.

**Methods And Evaluation Criteria:**

Given an unlabeled preference dataset, Sargy integrates AI-generated labels with selective human feedback to maximize alignment while minimizing annotation effort. The first stage is the initial alignment. A prompt is used to generate the preferences. The second is iterative alignment improvement, where bad labels are improved. A reward model is trained iteratively with selective human annotations to enhance alignment. Then the key relies in analyzing the distribution of the predicted reward function within the training preference dataset. By ranking all preference pairs, a monotonic reward distribution curve emerges - the upper left region shows high agreement between training data and the reward model, and high disagreement for the bottom right. The latter ones are used for annotation. However, in practice the ground-truth labels are unknown and the authors propose to use either the elbow or the knee. The human annotation beings from the reflection point. For the next iteration, the authors propose two techniques to combine the datasets: back-off ratio and amplification ratio.

**Other Comments Or Suggestions:**

N/A

**Other Strengths And Weaknesses:**

Overall, the paper is very well written, the methodology is sound, and the results good. My only criticism would be to include more baselines in the experiments, and increase the number of turns. Interestingly, it would be good to have so computational/cost analysis w.r.t. the baseline.

**Questions For Authors:**

How do the performance improve of 5 iterations?
What are the cost of the proposed method?

**Relation To Broader Scientific Literature:**

The method is novel.

**Theoretical Claims:**

N/A

---

> ### Author Rebuttal · Authors · 2025-04-01
>
> Thank you for your recognition and constructive review of our work!
>
> **Q1:** Other variants that are e.g., Greedy where the upper-left or bottom-right samples of the reward distribution are being annotated
>
> **Re:** This will indeed be an interesting factor to compare quantitatively in the final version. Given the time limit, we refer to Figure 2 and Figure 7 for a qualitative estimation. The accuracy curves show that:
> - For the "upper-left" area, most RM preferences are correct, and annotations do not introduce many changes to the next-iteration training
> - For the "bottom-right" area, <10% RM preferences are incorrect. Direct flipping is a more efficient way instead of annotation.
>
> **Q2:** Performance for more than 5 iterations
>
> **Re:** We have not included experiments after Itr-5 since the corresponding downstream LLM has already outperformed the one with full-human annotation and the human annotations have covered 20% of all samples. More annotations will gradually bring the RM closer to its counterpart in the full annotation setting (accuracy converges around the full-annotation accuracy).
>
> However, we have extended the experiments with 10 iterations under a 1/4 down-sampled shard of the full dataset (please kindly refer to the response to Q3 of Reviewer fzYM for more context). In each iteration, 4% of the subset (1% of the full set) receive human annotations. The test accuracy of the RM is listed as follows:
>
> | # Iteration | 1/4 Shard (HH-RLHF) | 1/4 Shard (TL;DR) |
> |-----|---------------------|------------------|
> | 0   | 78.4                | 81.0             |
> | 1   | 81.3                | 81.9             |
> | 2   | 84.6                | 83.8             |
> | 3   | 86.3                | 84.3             |
> | 4   | 87.6                | 85.2             |
> | 5   | 88.8                | 86.5             |
> | 6   | 89.6                | 87.4             |
> | 7   | 89.7                | 88.0             |
> | 8   | 90.3                | 88.3             |
> | 9   | 90.3                | 88.2             |
> | 10  | 90.8                | 88.4             |
>
> Note that here the size of the dataset is different from the one used in our original submission, therefore leading to different starting points and ceilings.
>
> We will also include results for more iterations across more sharding options in our final version.
>
> **Q3:** Computational/cost analysis w.r.t. the baseline
>
> **Re:** We have added a thorough computational/cost analysis. Please kindly refer to the response to Q3 of Reviewer fzYM.

---

> > ### Comment · Reviewer_BWsT · 2025-04-03
> >
> > Thank you for your answer. I will keep my score as is.

---

> > > ### Author Response · Authors · 2025-04-04
> > >
> > > Thank you again to all the reviewers for your constructive comments. We hope our responses have answered your questions and addressed your concerns.  With the discussion period ending in a few days, we wanted to emphasize that we are here to provide more information as you request.

---

### Official Review · Reviewer_fzYM · 2025-03-20

**Overall Recommendation:** 3

**Summary:**

The paper introduces Sargy, an iterative human-AI hybrid framework for aligning large language models (LLMs). The core idea is to leverage a reward model to identify data points that are difficult for an AI to label consistently with human preferences and then to selectively solicit human feedback on these challenging instances. By focusing human annotation efforts on these "hard-to-label" samples, the authors demonstrate that their approach achieves performance comparable to models trained with full human annotation, while using only 15-20% of the human annotation effort. This significantly reduces the cost associated with aligning LLMs.

## update after rebuttal
I have decided to maintain my original score. My assessment remains that the novelty of the proposed technique is borderline, as it bears significant resemblance to existing filtering methods.

**Claims And Evidence:**

In general, yes.

**Essential References Not Discussed:**

As I mentioned in method and evaluation criteria section. The author might need to discuss the literature on LLM training data filtering, an example:
[1]: RIP: Better Models by Survival of the Fittest Prompts

**Experimental Designs Or Analyses:**

See Methods And Evaluation Criteria

**Methods And Evaluation Criteria:**

The paper's approach of using the reward gap to identify and relabel hard-to-annotate preference pairs with human is interesting. The comparison with a random selection baseline effectively highlights the benefits of their targeted annotation strategy.

However, there are potential confounding factors to consider. A recent study [1] suggests that simply filtering out preference pairs with a large reward gap, regardless of the direction of the gap, can lead to significant performance improvements. This raises the question of whether the performance gains observed in this paper are solely due to the relabeling of the bottom percentile of the reward gap distribution, or if a similar improvement could be achieved by just filtering these instances without human intervention. The authors might consider exploring the impact of filtering and compare with relabeling to demonstrate the necessarily of human annotation.

Furthermore, the paper primarily focuses on the final aligned LLM's performance. It would be beneficial to also analyze the accuracy of the reward model itself. For instance, examining whether the iterative training process with the relabeling strategy leads to a demonstrably better reward model could provide valuable insights into the effectiveness of the proposed approach. Evaluating metrics specific to the reward model would help disentangle its improvement from other factors that might influence the final LLM's performance.

[1]: RIP: Better Models by Survival of the Fittest Prompts

**Other Comments Or Suggestions:**

NA

**Other Strengths And Weaknesses:**

Strengths:

The paper is well-written and easy to understand. The reported results, achieving near oracle-level performance with significantly reduced human annotation, are impressive and highlight the potential of the proposed framework.

Weaknesses:

As discussed in the previous section, the paper could benefit from a more in-depth analysis of the reward model's performance and a comparison with a simple filtering strategy based on the absolute reward gap, as suggested by recent work [1].

The paper should address the potential increase in training time due to the iterative nature of the Sargy framework. The process involves not only labeling data using an LLM judge but also training a reward model on the fly. This additional complexity might lead to a more cumbersome and time-consuming training pipeline compared to traditional methods. Quantifying this overhead and discussing potential optimizations would be valuable.

**Questions For Authors:**

NA

**Relation To Broader Scientific Literature:**

No

**Theoretical Claims:**

NA

---

> ### Author Rebuttal · Authors · 2025-04-01
>
> Thank you for your recognition and constructive review of our work!
>
> **Q1:** Comparison with a simple filtering strategy
>
> **Re:** Thanks for the relevant reference! The RIP paper was first published right around the submission deadline (1/30), and we will cite it in our final version. After careful reading, we want to highlight that:
> - RIP focuses on selecting more effective data while Sargy focuses on improving alignment with human preference. They can work together.
> - The improvement from human feedback is validated by our experiments: the ablation study (section 4.1.5) with "No Annotation" is exactly a representative of "filtering the bottom percentile of reward gap distribution". Sargy gets solid gain compared to this.
>
> **Q2:** Analysis of the accuracy of the reward model itself
>
> **Re:** The primary metric for all results in Section 4.1 is the preference accuracy of the RM itself. We will explicitly mention that in our final version.
>
> **Q3:** Quantify the training overhead
>
> **Re:** We take our experiments on HH-RLHF as a case study
> - **Dataset Size:** 160,800 samples, each with a prompt + 2 responses
> - **Human Annotation Cost:** Amazon Mechanical Turk [1] suggested text classification pricing: 0.012 * 3 (labelers) = \\$0.036 per sample
>
>   Note: Here the suggested pricing may be much lower than the actual cost. Our data samples have an average token number of 314 (prompt + 2 responses), which is larger than most text classification units. AMT's labeling service providers typically list an hourly rate of \\$6-7. According to human reading speed of 200-250 words per minute, the actual cost should be around \\$0.13-0.18/sample/labeler, which is more than 10x of the suggested pricing. In the following analysis, we still use the suggested pricing as a lower bound to provide a conservative estimate of Sargy's gain.
>
> - **LLM Annotation Cost:**
>   - Average input length (template + prompt + 2 responses): 671 tokens
>   - Average output length (rational + judgement): 134 tokens
>   - OpenAI API cost (per 1M tokens)
>
>     **GPT-4o:** \\$2.5 for input; \\$10 for output
>     - 671 * 0.0000025 + 134 * 0.000010 = \\$0.0030 per sample
>
>     **GPT-4o mini:** \\$0.15 for input; \\$0.6 for output
>     - 671 * 0.00000015 + 134 * 0.0000006 = \\$0.00018 per sample
> - **RM Training & Inference Cost:** Azure ML costs \\$32.77 per hour for a 8xA100 80GB node [2]. A Sargy RM training + inference per iteration takes less than 8 hours on the full dataset, and less than 2 hours on the 1/4 subset. The inference time is negligible compared to training time.
> - **Comparison:** (For computing, we only consider RM training + inference, as the downstream LLM training is the same for both full-human annotation and Sargy)
>   - **a. Full-human annotation:** 0.036 * 160800 (human annotation cost) = \\$5788.8
>   - **b. Sargy (full set + GPT-4o + 0-5 iterations) + 20% human annotation:** 0.0030 * 160800 (LLM annotation) + 0.036 * 160800 * 20% (human annotation) + 32.77 * 8 * 6 (training + inference) = \\$3213.1
>   - **c. Sargy (full set + GPT-4o mini + 0-5 iterations) + 20% human annotation:** 0.00018 * 160800 (LLM annotation) + 0.036 * 160800 * 20% (human annotation) + 32.77 * 8 * 6 (training + inference) = \\$2759.7
>
>   Our additional experiments on both TL;DR and HH-RLHF show that by down-sampling the full dataset to a 1/4 shard for Sargy’s processing and conducting inference (scoring) on the full dataset **only at the end** using the reward model from the final iteration, we achieve accuracy comparable to using the full dataset throughout Sargy's process. This approach not only reduces computational costs but also decreases the required human annotations **(6-7% instead of 15-20%)** w.r.t. to the full dataset. We will include these results in the final version. With such approach, the cost of Sargy can be further reduced:
>   - **d. Sargy (1/4 shard + GPT-4o + 0-6 iterations) + 6% human annotation:** 0.0030 * 160800 * 1/4 (LLM annotation) + 0.036 * 160800 * 6% (human annotation) + 32.77 * 2 * 7 (training + inference) = \\$926.7
>   - **e. Sargy (1/4 shard + GPT-4o mini + 0-6 iterations) + 6% human annotation:** 0.00018 * 160800 * 1/4 (LLM annotation) + 0.036 * 160800 * 6% (human annotation) + 32.77 * 2 * 7 (training + inference) = \\$813.3
>
>   Even counting the extra LLM labeling and computing overhead, Sargy can still reduce the overall cost by **44.5-86.0%**. Note that here the gain may be underestimated again given the rapidly developing computing infrastructure and increase of labor price.
>
>   We will include this analysis in our final version.
>
> [1] https://aws.amazon.com/sagemaker-ai/groundtruth/pricing/
>
> [2] https://learn.microsoft.com/en-us/azure/virtual-machines/sizes/gpu-accelerated/ndma100v4-series?tabs=sizebasic

---

### Official Review · Reviewer_7upR · 2025-03-25

**Overall Recommendation:** 3

**Summary:**

This paper introduces Sargy, a human-AI hybrid framework designed to improve LLM alignment with user preferences while minimizing human annotation costs. Sargy strategically combines LLM-generated labels with selective human corrections, identifying and refining mislabeled samples using a reward model’s distribution. The framework operates in three stages: (1) Initial alignment, where an LLM provides coarse labeling; (2) Iterative improvement, leveraging human feedback to correct challenging samples; and (3) Knowledge transfer, using the refined dataset for downstream preference optimization tasks like DPO and PPO. Experiments on HH-RLHF and TL;DR datasets demonstrate that Sargy achieves oracle-level alignment with just 15–20% of the human annotation effort, enabling high-quality preference learning with minimal cost.

**Claims And Evidence:**

The paper claims that Sargy can achieve oracle-level alignment while reducing human annotation effort to just 15–20% of the full dataset. The experimental results on HH-RLHF and TL;DR datasets support this claim, showing that models trained on Sargy’s filtered datasets perform on par with those trained on fully annotated data. However, the effectiveness of this reduction likely depends on dataset difficulty, which is not fully explored in the paper. Additionally, since reward models may rely on spurious correlations, their alignment scores do not always guarantee correct annotations, raising concerns about robustness.

**Essential References Not Discussed:**

I did not find any missing essential references.

**Experimental Designs Or Analyses:**

The experiments demonstrate that Sargy effectively reduces annotation costs while maintaining alignment quality. However, an ablation study on dataset difficulty and reward model stability is missing.

**Methods And Evaluation Criteria:**

The paper presents a three-stage framework that combines LLM-generated labels with selective human feedback, guided by a reward model’s reward distribution. The evaluation is conducted on preference datasets (HH-RLHF and TL;DR) and assesses alignment quality through downstream task performance. While the method is straightforward and practical, a more detailed discussion of dataset complexity and reward model stability would strengthen the evaluation.

**Other Comments Or Suggestions:**

Please see my comments for other questions.

**Other Strengths And Weaknesses:**

Please see my comments for other questions.

**Questions For Authors:**

1. The paper claims that using 15–20% of the data achieves performance comparable to using the full dataset. How does this generalize to more complex datasets?

2. What happens if the reward model predicts that most of the data is misaligned with human preferences?

3. Since reward models may rely on spurious correlations, how does the method ensure the stability and reliability of reward alignment for guiding human annotation?

4. How sensitive is the framework to the initial LLM-generated labels? Would errors in this stage propagate and affect the final performance?

**Relation To Broader Scientific Literature:**

The paper aligns with research on RLHF, human-AI collaboration in annotation, and active learning.

**Theoretical Claims:**

The paper does not introduce new theoretical claims.

---

> ### Author Rebuttal · Authors · 2025-04-01
>
> Thank you for your recognition and constructive review of our work!
>
> **Q1 - Data complexity:** The paper claims that using 15–20% of the data achieves performance comparable to using the full dataset. How does this generalize to more complex datasets?
>
> **Re:** The generalizability of Sargy across complex tasks is indeed a genuine point that we also tried to investigate in the paper. Recent research [1] suggests that the complexity of a task is a function of the model capability. To address this, we intentionally use GPT-4o mini as a representative of comparatively weaker models for initial feedback, effectively making the task harder. Our experiments in 4.1.1 show that even when starting with a weaker model (i.e., a harder task), Sargy consistently closes the gap with stronger initial models (GPT-4o) and achieves **similar preference accuracy** after 5 iterations with equal amount of data and human annotation.
>
> **Q2 - Major RM misalignment:** What happens if the reward model predicts that most of the data is misaligned with human preferences?
>
> **Re:** To address this pragmatic consideration, Sargy natively incorporates a validation process of initial alignment (Section 3.2.3 paragraph-1). If a major misalignment is found, the initial alignment prompt needs to be updated according to the misaligned samples / users / any existing techniques. We will further highlight that in our final version.
>
> **Q3 - Reward model stability:** Since reward models may rely on spurious correlations, how does the method ensure the stability and reliability of reward alignment for guiding human annotation?
>
> **Re:** Sargy provides a generic framework that can correct mistakes for any reward models. Recent evidence [2] has shown that stronger base models may better capture the real target correlations during reward modeling. In our experiments, we found Llama-3.1-8B-Instruct on par in aligning with human preference.
>
> **Q4 - Initial alignment quality:** How sensitive is the framework to the initial LLM-generated labels? Would errors in this stage propagate and affect the final performance?
>
> **Re:** Sargy’s robustness to initial alignment quality is a key strength of our approach. (Similar to Q1) Experiments in section 4.1.1 show that even when starting with a poorly aligned model (GPT-4o mini), Sargy consistently **closes the performance gap with stronger models** after the **same number of human annotations and iterations**. A key reason for this is Sargy's strategic data selection: when initial alignment is poor, the selected samples for human annotation tend to have a higher proportion of mislabeled instances. This targeted selection ensures that human feedback corrects a larger number of errors per annotation, leading to a **higher improvement-per-annotation ratio**.
>
> [1] Scaling LLM Test-Time Compute Optimally can be More Effective than Scaling Model Parameters
>
> [2] Scaling Laws for Reward Model Overoptimization

---

### Decision · Program_Chairs · 2025-05-01

**Decision:**

Accept (poster)

**Comment:**

My recommendation is to accept the paper.

The paper proposes Sargy, a workflow for augmenting LLM-labeled preference datasets with human annotations to obtain similar performance to using pure human annotation with substantially reduced human annotation effort. The process iteratively updates a reward model, by filtering data and obtaining additional human annotations. Reviewers generally agreed that the results were compelling. There were some questions raised about novelty wrt filtering methods, but the authors responded adequately to these questions.

Having looked over the paper myself, I would suggest that the authors include a succinct algorithm-box-like summary of the iterative procedure, particularly the construction of the score gap ranking curve at each step, as there is currently only long-form description of the steps. This would make it easier to understand the procedure and replicate the results. In particular, it seems like it would be easy to leak information from ground truth ranking labels (through the sign of the gap) if this construction is done incorrectly by readers.